# Mitotic Functions and Characters of KIF11 in Cancers

**DOI:** 10.3390/biom14040386

**Published:** 2024-03-22

**Authors:** Wanting Gao, Junjie Lu, Zitao Yang, Enmin Li, Yufei Cao, Lei Xie

**Affiliations:** Department of Biochemistry and Molecular Biology, Shantou University Medical College, Shantou 515041, China; 20wtgao@stu.edu.cn (W.G.); 20jjlu@stu.edu.cn (J.L.); 22ztyang@stu.edu.cn (Z.Y.); nmli@stu.edu.cn (E.L.)

**Keywords:** KIF11, mitosis, cancer, epithelial–mesenchymal transition, angiogenesis

## Abstract

Mitosis mediates the accurate separation of daughter cells, and abnormalities are closely related to cancer progression. KIF11, a member of the kinesin family, plays a vital role in the formation and maintenance of the mitotic spindle. Recently, an increasing quantity of data have demonstrated the upregulated expression of KIF11 in various cancers, promoting the emergence and progression of cancers. This suggests the great potential of KIF11 as a prognostic biomarker and therapeutic target. However, the molecular mechanisms of KIF11 in cancers have not been systematically summarized. Therefore, we first discuss the functions of the protein encoded by KIF11 during mitosis and connect the abnormal expression of KIF11 with its clinical significance. Then, we elucidate the mechanism of KIF11 to promote various hallmarks of cancers. Finally, we provide an overview of KIF11 inhibitors and outline areas for future work.

## 1. Introduction

Kinesin family member 11 (KIF11/Kinesin-5/Eg5/BinC/KSP) is the first spindle motor to be identified through temperature-sensitive fungal library screening, named BimC in *A. nidulans* and Cut7 in yeast [1,2]. Subsequent studies have identified BimC/Cut7 orthologs, such as Eg5 in *Xenopus* and hsEg5, in humans [3]. Functioning in spindle assembly, this gene is named KIF11 and classified within the Kinesin-5 family [4]. Shown in the National Library of Medicine, human KIF11 is located in band q23.33 of chromosome 10 and is composed of 1057 amino acids. As the amino terminal region contains a motor domain, KIF11 belongs to the N-terminal kinesins [5]. The intact KIF11 product is a bipolar homotetramer with three domains contributing to kinesis function differentiation. The N terminal motor domain is responsible for binding microtubules (MTs) and hydrolyzing ATP, acting in a two-heads-bound state to move the MTs [6]. The coiled coils of the tetramer are formed by the stalk domain. Furthermore, the tail domain plays a role in subcellular localization, and its variations determine the pattern of binding specific cargo [7,8]. The motor domain of KIF11 consists of several conservative structures, including a core structure and three essential conformational changing structures (Figure 1). It has a central β-sheet with α-helices at each end and contains loop 5 [9,10], switch I [11], and switch II [11,12]. The nucleotide induces the conformational change in these three structures to transmit information to the neck linker and MT-binding sites, directly triggering movement and cooperation between four motor domains, which was first found in the Kinesin-1 family [10,13,14]. Exhibiting the KIF11 structures, X-ray crystallography provides opportunities to develop small-molecule inhibitors.

KIF11 forms and maintains the bipolar spindle during mitosis [15,16]. Based on its indispensable contributions to accurate chromosome segregation, numerous studies have elucidated the crucial involvement of KIF11 in cancer development. Upregulated KIF11 has been identified in various cancers, such as in glioma [17], adrenocortical carcinoma (ACC) [18], colorectal cancer (CRC) [19,20], non-small-cell lung cancer (NSCLC) [21,22], and hepatocellular carcinoma (HCC), and is related to poor prognosis [23,24,25]. KIF11 promotes cancer proliferation, invasion, and metastasis mainly through regulating the cell signaling pathways [26,27]. Additionally, KIF11 is regulated at the transcriptional, post-transcriptional, and post-translational levels in tumorigenesis and tumor progression [19,28,29,30,31,32]. However, a comprehensive summary of these essential roles is lacking. Notably, having been identified as a promising target for therapeutic interventions, KIF11 attracts widespread interest in this field. Compared with paclitaxel, a taxane belonging to a classical anti-mitotic drug targeting tubulin, KIF11 inhibitors induce mitotic arrest and aberrant monoastral without significant neurotoxicity due to their low-level expression in neurons [33,34,35,36,37,38,39,40,41]. However, in another exploration of the expression and role of KIF11 in neurons, neurotoxic effects have also been reported after treating brain tumors [42]. The potential risk of complications with learning ability and memory after using KIF11 inhibitors cannot be ignored [43].

Hence, herein, we identify and discuss the mitotic functions of KIF11, paying attention to the accessory proteins. Then, we summarize the clinical significance and regulation of KIF11 expression, present the oncogenic functions of KIF11 in different hallmarks of cancers, and, finally, summarize the research on KIF11 inhibitors and offer future research directions.

## 2. Functions of KIF11 during Mitosis

### 2.1. KIF11 in Centrosome Events

Being an MT-organizing center, the centrosome contributes to correct MT localization and mitotic spindle assembly [44]. The orientation of bipolar spindles in *D. melanogaster* male germ stem cells requires the centrosome to function normally, which is mediated by the accurate localization of KIF11 [45,46]. KIF11 has been reported to be present in the cytoplasm during interphase, targeting the centrosome and spindle pole in prophase. Subsequently, plus-end-directed motility propels KIF11 to localize on the spindle MTs during metaphase [47,48]. Mediated by the kinase cascade reaction, the accurate positioning of KIF11 is controlled by the mitotic kinases cyclin-dependent kinase 1 (CDK1), polo-like kinase 1 (PLK1), and aurora kinase A (AURKA) (Figure 2). CDK1 directly phosphorylates KIF11 at T926 to promote its accumulation in the centrosome [49]. In addition, CDK1 and PLK1 synergistically promote the phosphorylation of NIMA-related kinase (NEK) 9 to activate NEK6/7, consequently inducing the phosphorylation of KIF11 at S1033 to control prophase centrosome separation [50,51,52]. Controlled by the NEK9/6/7 cascade, discs large homolog 1 (DLG1) partners with KIF11 and colocalizes at the centrosome [53,54]. This docking relies on the PDZ binding domain of the phosphatase and tensin homolog (PTEN) [53,55]. Similar localization regulated by RAB11 is facilitated via other mitotic regulators rather than directly binding with KIF11 [56]. Additionally, PLK1 is activated by AURKA, involving the centrosome and spindle pole localization of KIF11 [57,58]. The silencing of IκB kinase 2 (IKK2) results in the stabilization of AURKA and an increase in the phosphorylated KIF11 level, leading to the formation of a multipolar spindle [59].

KIF11 is the main force generator in centrosome separation, which promotes bipolar spindle formation. Inhibiting focal adhesion kinase (FAK) represses the phosphorylation of PLK1 and KIF11, leading to failed centrosome separation [60]. Mardin et al. suggested that premature centrosome separation occurs when the overexpression of either NEK9 or NEK6 increases the motor force provided by KIF11 [61]. Additionally, Eibes et al. found that, guaranteed by S1033 phosphorylation of KIF11, the extreme C terminus of the targeting protein for *Xenopus* kinesin-like protein 2 (TPX2) directly interacts with the tail domain of KIF11, inducing the assembly of KIF11 on the centrosome to execute centrosome segregation [62]. TIAM1 antagonizes KIF11-inducing centrosome segregation via phosphorylation by CDK, causing the p21-activated kinases (PAKs) 1/2 to act on RAC. These results further indicate that downregulated PAK1/2 allows cancer cells to escape from the unipolar inhibition induced by KIF11 inhibitors, providing the theoretical basis for the study of resistance to KIF11 inhibitors [63].

Centrosome amplification (CA), a preventive behavior of cancer cells, gathers excess centrosomes to prevent harmful multipolar division, which is also crucial for normal cells living with CA [64,65,66]. Drosopoulos et al. demonstrated that the stability and protein level of KIF11 are increased by the inhibition of anaphase-promoting complex/cyclosome (APC/C), breaking the force balance on the mitotic spindle, halting CA, and resulting in multipolar spindle formation as well [67].

### 2.2. KIF11 in Spindle Events

The prominent function of KIF11 is crosslinking and sliding anti-parallel MTs apart to build bipolar spindles via plus-end-directed motility during mitotic spindle construction. The motion domain has been identified as the structural basis promoting this process [68]. However, Li Tao et al. found that both the motor and tail domains of KLP61F, KIF11 in *Drosophila melanogaster*, can crosslink MTs in the presence of the stalk domain [69]. Moreover, by facilitating the interaction between KIF11 and MTs, the tail domain promotes their crosslinking and sliding to ensure the effective alignment of MTs during metaphase without substantially hindering movement [70]. The tail domain interacts with the motor to decelerate ATP hydrolysis, generating a stable, powerful force to confirm the crosslinking and sliding of MTs [71,72].

Controlled by the ATP cycle, slow and plus-end-directed motions of KIF11 have been demonstrated by Kapitein et al. [73]. The push or pull force generated by directed motion might be positively proportional to the geometric features of the dynamic MT network, such as the overlap length between adjacent filaments [74]. Further evidence establishes a linear correlation between the force exerted on the overlapped anti-parallel MTs and the quantity of motor proteins, as well as the length of MTs’ overlaps. Meanwhile, a kind of brake-like force can be generated by KIF11 to withstand the sliding motion of filaments in both parallel and anti-parallel structures [75]. Surprisingly, the minus-end-directed motion of cin8, the kinsin-5 motor of yeast, occurs when either cin8 acts in a low MT protein concentration or on individual MT singly. In contrast, the direction alternates to the plus end when cin8 is in a high motor protein concentration or works as a group of motors on anti-parallel MTs [76]. The bi-directional mobility could be explained by the fact that the motion domain contains one or more nonclassical MT-binding sites, indicating that each MT dimer binds 3–5 cin8 motors [77]. However, the role of minus-end-directed movement remains unclear. Gergely et al. found that the minus-end-directed motion promotes the motility of the motor and positions cin8 at the spindle pole, contributing to MT crosslinking as well as the separation of spindle poles [68,78]. Bi-directional movement has not been identified in other kinesin-5 proteins. Hence, it is still unclear whether this is a unique mechanism in yeast cells or a common feature of kinesin 5.

KIF11 is an important regulator of mitotic spindle dynamics. Cin8p and Kip1p, KIF11 homologs found in budding yeast, have been identified as depolymerases of MTs. Other studies have shown that the length-dependent regulation of kinesin-5 promotes the disassembly of net kinetochore MTs [79]. Mechanically, the stability of the lattice is destroyed because of increasing stress on tubulin–tubulin bindings caused by the mechanical force between stepping motor domains. Alternatively, KIF11 may carry cargo to facilitate the net disassembly of the plus end of MTs. Nucleophosmin/B23 is the cargo of KIF11 in HeLa cells, and the DNA/RNA-binding domain of B23 binds to the KIF11 motor domain, inhibiting ATPase-induced MT disassembly and mutation [80]. However, in other studies on budding and fission yeasts, KIF11 mainly shows MT polymerase rather than depolymerase activity [81,82]. Single-molecule fluorescence experiments in vitro revealed the accumulation of KIF11 dimers in plus-ended MTs, promoting MT growth [15]. Further evidence suggests that the crosslinking of the dimer motor of KIF11 is unnecessary for promoting MT formation, while a high concentration of KIF11 monomers can also promote MT construction. However, the KIF11 dimer plays a more efficient role in promoting MT assembly, stabilizing MTs and halting tubulin dissociation, thereby enhancing lattice stability [83].

### 2.3. Binding Partners of KIF11 in Spindle Events

The nuclear mitotic apparatus protein (NuMA) colocalizes with KIF11 in the spindle pole, mediating mitosis together. Further results reveal that KIF11 directly interacts with NuMA via the stalk domain, thereby regulating the spindle pole localization of NuMA. However, NuMA does not have a regulatory effect on KIF11 localization [84]. In contrast with NuMA, dynein contributes to the accumulation of KIF11 in the spindle pole, which has been identified in both *Xenopus laevis* eggs and intact mammalian cells [85,86]. However, the depletion of dynein in HeLa cells has no significant effect on the spindle pole localization of KIF11. Therefore, the regulatory mechanism of KIF11 localization varies among different cells and tissues [84]. Described as a simple push-pull model, dynein antagonizes KIF11 to pull on the MTs, promoting the collapse of the spindle. However, subsequent research suggests that dynein utilizes force in various spindle positions to indirectly antagonize KIF11 [87]. Dynein independently separates the spindle poles or collaborates with KIF11 during prophase. It also regulates the pole-directed movement of KIF11, influencing the spindle length and structure during metaphase [88]. Meanwhile, dynein facilitates the fusion of spindles in KIF11-inhibited *Xenopus* eggs during meiosis II to form monopolar spindles through the opposite directional movement of MTs [89].

TPX2, a microtubule-associated protein, interacts with KIF11 to form a complex via the C-terminal 35 residues of TPX2 [90,91]. The TPX2-KIF11 complex localizes in the spindle pole, regulated by vacuolar protein sorting-associated protein 28 (VPS28), an ESCRT-I protein [92]. A high level of TPX2 recruits KIF11 to the spindle polar, regulating its structure [93]. But the distribution pattern of these two proteins is not completely uniform during mitosis. Though both TPX2 and KIF11 target the spindle pole in metaphase, the accumulation of TPX2 is two-fold more than KIF11 enrichment. During anaphase, KIF11 redistributes to the spindle midzone, while most TPX2 still accumulate in the spindle pole [94]. These results indicate that the regulation of KIF11 and TPX2 might have common and independent approaches. It is reported that the TPX2-KIF11 complex plays an important role in maintaining the bipolar spindle and balancing the force between poles [95,96]. TPX2 could reduce KIF11 velocity using two models. One is a combination between TPX2 and the MT lattice, which is independent of the C-terminal 35 residues of TPX2, and another involves TPX2 directly interacting with KIF11. Additionally, full-length TPX2 significantly diminishes the MT gliding ability of the KIF11 dimer. In contrast, truncating the binding domain of TPX2 to interrupt the interaction with KIF11 has a smaller inhibitory effect. However, two types of TPX2 do not have a significant effect on the monomer KIF11 [95]. It is speculated that the KIF11 stalk domain is essential to this different inhibiting function and may be required for combining with TPX2, but the fifty percent reduction in the rate of the MT gliding of the KIF11 monomer compared with that of the dimer may also cause this difference [93]. Heat shock protein 70 (HSP70) is a molecular chaperone and regulates the interaction between TPX2 and KIF11. Silencing HSP70 stabilizes the TPX2-KIF11 complex at the spindle pole and represses the oligomerization of KIF11, preventing the inward movement of KIF11 during metaphase [97].

## 3. Clinical Relevance of Dysregulated KIF11

Aberrant upregulated KIF11 in most cancer types has been demonstrated via pan-cancer analysis and immunohistochemistry-based analyses, as well as in vitro experiments, contributing to tumorigenesis and progression [98] (Table 1). Surprisingly, a study on the immunohistochemical expression of KIF11 in pancreatic ductal adenocarcinoma (PDAC) tissues conducted by Klimaszewska-Wiśniewska et al. showed high-intensity but less-abundant KIF11 staining, and the Kaplan-Meier curves suggested that the PDCA patients with overexpressed KIF11 had better overall survival (OS) rates [99]. Conflicting findings provided by M. Liu and X. Gu indicated that KIF11 is upregulated in PDCA and is associated with a worse OS rate [100,101]. These results imply a potentially intricate expression pattern for KIF11 in PDCA. Various clinicopathological features (e.g., tumor size and pathological stage and TNM stages) are significantly associated with overexpressed KIF11, which has been revealed in a series of cancers types, such as ACC [18], NSCLC [102,103], CRC [19,104], prostate cancer (PCa) [105], and breast cancer [106], among others. It seems that KIF11 plays more important roles in pathogenesis rather than progression in esophageal squamous cell carcinoma (ESCC), since only the N classification is related to the upregulated KIF11, which has no significant association with survival [107].

Studies focusing on the roles of KIF11 in the tumor microenvironment (TME) are relatively limited. Acting as one of the essential features of the TME, immune cell infiltration related to KIF11 has been reported in ACC and lung adenocarcinoma (LUAD). Li et al. revealed that a high expression level of KIF11 partially increases the abundance of B cells and dendritic cells [108]. Another Spearman correlation analysis revealed that KIF11 expression is positively correlated with the Th2 cell count and negatively correlated with CD8+ and mast cell counts, among others [18]. Because of the better prognosis predicted by the immune cell group in ACC, the roles of KIF11 in immune cell infiltration need to be explored [109]. Similar disparate roles of KIF11 have been exhibited in LUAD patients. Predicting a good prognosis, the overexpression of KIF11 is negatively associated with the infiltration of resting memory CD4 cells and monocytes, whereas the negative correlation with T regulatory cells indicates poor a prognosis [110]. Similarly, a positive relationship between KIF11 expression and natural killer cells predicts a poor prognosis as well [110]. Meanwhile, immune cells interact with normal cells to impact the metabolism in the TME, promoting cancer and lipid metabolism involving KIF11, causing PDAC progression [101,111]. This mechanism is described in detail below, but the potential association with immune regulation in PDAC must be investigated in the future. Overall, even though the role of KIF11 in immune cell infiltration needs to be researched, these studies provide insight into immune therapy strategies based on the dysregulation of KIF11.

**Table 1 biomolecules-14-00386-t001:** Clinical relationships of abnormal KIF11 expression.

Cancer Types	Clinical Relationships	Ref.
Glioma	Upregulated KIF11 is negatively associated with overall survival (OS) and increased the chemotherapy resistance to 5-FU in glioma cells.	[28]
Glioblastoma	Upregulated KIF11 is positively related to the WHO grade and poor OS in high-grade glioma. Preclinical models show that inhibiting KIF11 prolongs tumor latency and survival.	[17]
Astrocytoma	Upregulated KIF11 is positively related to the WHO grade of astrocytoma, especially grading astrocytic tumors.	[112]
Adrenocortical carcinoma	Upregulated KIF11 is related to poor OS and positively related to immune infiltration, tumor size, T stage, pathological stages, and tumor recurrence rate within 2 years.	[18,108]
Meningioma	Upregulated KIF11 is positively related to the WHO grade of meningioma, as well as shorter progression-free survival (PFS).	[113]
Non-small-cell lung cancer	Upregulated KIF11 is positively related to pathological stages, T stage, and the occurrence of lymph node metastases; upregulated KIF11 is associated with poor OS and is positively related to lymph node metastases and pathological stages. Upregulated KIF11 is associated with poor disease-free survival (DFS).	[21,102,103]
Lung adenocarcinoma	KIF11 is involved in the cell cycle, tumor microenvironment alternation, and tumor-infiltrating immune cell proportions; upregulated KIF11 results in higher infiltration levels.	[110,114]
Laryngeal squamous cell carcinoma	Upregulated KIF11 predicts poor prognosis and is positively related to lymph node metastasis and TNM stages.	[115]
Hepatocellular carcinoma	High expression of KIF11 exhibits poor OS and DFS.	[23]
Colorectal cancer	High expression of KIF11 is significantly associated with the lymphovascular invasion; high expression of KIF11 predicts poor OS; overexpression of KIF11 is significantly related to T stage, M stage, and TNM stages and vessel invasion; overexpression of KIF11 is associated with T classification and well or moderate tumor differentiation.	[19,104,107,116]
Esophageal squamous cell carcinoma	Overexpression of KIF11 is associated with N classification.	[107]
Pancreatic ductal adenocarcinoma (PDAC)	Overexpression of KIF11 is related to poor OS and positively associated with tumor grades, lymphatic metastasis, and clinicopathological stages; the KIF11 protein is downregulated in PDAC specimens, whereas KIF11 mRNA is upregulated. Both a low level of KIF11 protein and high level of KIF11 mRNA predict poor OS.	[99,101]
Bladder cancer	Overexpression of KIF11 is associated with tumor grade and poor OS.	[117]
Clear cell renal cell carcinoma	Overexpression of KIF11 is associated with tumor nuclear grade, TNM stages, and poor OS.	[118]
Cervical cancer	KIF11 is one of the prognostic factors.	[119]
Serous ovarian cancer	Overexpression of KIF11 is positively related to tumor grade, TNM stages, and lymph node metastasis.	[120,121]
Ovarian cancer	Overexpression of KIF11 is related to poor OS and PFS	[122]
Prostate cancer	Overexpression of KIF11 is related to higher T stage, poor MFS, and bone metastasis occurrence; KIF11 mRNA is upregulated in stage ≥ T3 or metastatic disease compared with T2N0M0 disease; overexpressed nuclear KIF11 is observed in T4 tumors and metastatic disease.	[105,123,124]
Breast cancer	Overexpression of KIF11 is associated with poor OS and high recurrence rates; overexpression of KIF11 is related to clinical stage, T/N/M classification, and poor survival time.	[106,125]

## 4. Regulation of KIF11

An accumulating body of evidence demonstrates the clinical implications of dysregulated KIF11 expression, driving extensive exploration into multiple functions of KIF11 in different cancers (Table 2); thus, it is of great importance to elucidate the regulation mechanisms underlying aberrant KIF11 expression at the DNA, RNA, and protein levels in various cancers (Figure 3).

### 4.1. Transcriptional Regulation of KIF11

In glioma, KIF11 is upregulated via the mutation of p53, which is a well-known tumorigenesis participator in many kinds of cancers, to promote the stemness, proliferation, and drug resistance of glioma cells [28,136]. However, further investigations are required to elucidate the mechanism between KIF11 and p53. The upregulation of KIF11 in gallbladder cancer (GBC) cells is associated with the enrichment of histone H3 lysine 27 acetylation (H3K27ac) in the promoter region [27]. The knockdown of K (lysine) acetyltransferase 5 (KAT5) has been identified to repress this enrichment, resulting in the autophagy and apoptosis of anaplastic thyroid carcinoma (ATC) cells [29]. Additionally, parkin, an E3 ligase, is found to downregulate the transcriptional level of KIF11. Mechanically, parkin deactivates the JNK/c-Jun pathway through the multiple monoubiquitination of HSP70, leading to decreased interaction between c-Jun and the KIF11 promotor [137]. In pancreatic adenocarcinoma, the downregulation of parkin stimulates the formation of spindle multipolarity, cancer cell proliferation, and tumorigenic properties by elevating the KIF11 expression level [138].

### 4.2. Post-Тranscriptional Regulation of KIF11

Various non-coding RNAs regulate the expression of KIF11. MicroRNA (miRNA) influences protein synthesis via the post-transcriptional regulation of messenger RNA (mRNA), which has been demonstrated to regulate KIF11 expression in cancers [139,140]. For instance, miR-186-5p binds to the 3′UTR of KIF11, inhibiting the proliferation of neuroblastoma [30]. In breast cancer, negative regulation between KIF11 and miR-30a has been elucidated [31]. In addition, KIF11 has been found to be the target of miR-424 and miR-381, playing roles in ovarian cancer [133]. Liu JF et al. confirmed that the long non-coding RNA VPS9D1-AS1 upregulates KIF11 expression via the competitive inhibition of miR-30a-5p in LUAD [141]. Interestingly, Liu YX et al. found that tumor necrosis factor receptor-associated factor 4 (TRAF4) may activate KIF11 translation. CircularRNA MTO1 interacts with TRAF4 and blocks the interaction between TRAF4 and KIF11. Consequently, the protein level of KIF11 is decreased to suppress the viability of breast cancer cells [142].

In addition, insulin-like growth factor-2 mRNA-binding protein 3 (IMP3) has been identified to conduct post-transcriptional regulation by binding to mRNA and promote the migration of malignant tumors [143,144,145]. Hence, Pasiliao et al. suggested that silencing IMP3 decreases the KIF11 mRNA level, thus inhibiting the migration, invasion, and adhesion of pancreatic cancer cells through alternating the cytoskeletal dynamics [146].

### 4.3. Post-Translational Regulation of KIF11

Various types of KIF11 post-translational regulation are essential in both normal cells and oncogenic functions in cancer cells, including phosphorylation, acetylation, and ubiquitination. The phosphorylation of KIF11 promotes its positioning at the centrosome and spindle, as well as the formation and maintenance of the spindle [52,62]. The functions and the specific phosphorylation sites have been summarized by Mann et al. [147] and Wojcik et al. [8]. NEK9 activates NEK6/7 and phosphorylates KIF11, playing an important role in spindle assembly, whereas an upregulated NEK9/KIF11 axis is related to the metastatic potential of colon cancer cells [104,148]. Cyclin A2 activates CDK1, which is an essential kinase for KIF11 phosphorylation, contributing to cell cycle regulation [149]. A low expression level of cyclin A2 halts KIF11 phosphorylation at T927 and causes spindle geometry defects in colon cancer [132]. The phosphorylation of KIF11 is becoming a research hotspot, while the dephosphorylation regulation of KIF11 is still unclear. Protein phosphatase 2A (PP2A) binds to the tail of KIF11 and dephosphorylates KIF11 at T926 [150]. The knockdown of PP2A hyperphosphorylates KIF11, resulting in delayed metaphase initiation and chromosome mis-segregation [150]. Furthermore, PTEN interacts with the motor and tail domains of KIF11, thus decreasing the phosphorylation level at T926 [55]. Induced by PTEN depletion, the excessive phosphorylation of KIF11 causes a morphological change in the spindle and relocalization of phosphorylated KIF11 [55]. A chromosome segregation defect can be induced by many events, such as spindle geometry defects and DNA double-strand breaks [151]. Notably, DNA double-strand breaks dephosphorylate cin8 in the telophase of mitosis and relocalize cin8 to partially reverse chromosome segregation [152]. However, the phosphorylation site of this process has not been determined yet.

Except for phosphorylation, the acetylation of KIF11, altering its movement, has been observed in several studies. K146 of KIF11 was found to be acetylated in a global acetylation proteomics study, and further results show that K146 acetylation slows spindle polar separation [153,154]. Nevertheless, Dhanusha et al. found that K890 is the main acetylation site of KIF11 rather than K146; acetylation also acts as a “break” to slow mitosis through decreasing the ATPase activity level of KIF11 [155]. Additionally, histone deacetylase 1 (HDAC1), an epigenetic enzyme, is observed to deacetylate KIF11 at K890, enhancing ATPase and completing the prophase events [155]. Moreover, Zheng JJ et al. claimed that N-acetyltransferase 10 (NAT10) regulates the mobility and targeting of KIF11 at the centrosome through acetylating KIF11 at K771, and the inhibition of NAT10 reduces KIF11 loading on the centrosome, leading to mitotic catastrophes [156].

Several proteins have been found to ubiquitinylate KIF11. CDH1 is an APC/C cofactor that induces the ubiquitination of KIF11 to maintain genomic stability. The mutagenesis of either the KEN box (aa 1022–1024) or two D box sequences (aa 944–947 and 1047–1050) of KIF11 results in the inability to bind to CDH1, leading to failed ubiquitination and degradation [157]. Additionally, in glioblastoma tumor-initiating cells that perform angiogenesis and invasion, APC/C^CDH1^ has been confirmed to be defective, leading to the stabilization of KIF11 throughout the cell cycle [17]. In ATC, the abnormal spindle-like microcephaly-associated protein (ASPM) is highly expressed and inhibits the ubiquitination of KIF11 to promote the movement of ATC cells [32]. In breast cancer, the RNF20/40 ubiquitin ligase complex monoubiquitinates and stabilizes KIF11 at K745 to promote breast carcinogenesis [158]. Additionally, TRAF4 decreases the level of SMAD-specific E3 ubiquitin protein ligase family member 2 (SMURF2) and inhibits the polyubiquitination of KIF11 to promote breast cancer cell proliferation [159,160]. Furthermore, F-box protein 30 (FBXO30) interacts with the tail domain of KIF11 and functions as the E3 ligase to promote the ubiquitination of KIF11 at K891 and K899. FBXO30 deficiency dysregulates KIF11 and causes mammopoiesis defects [161]. Considering the identification of the upregulated KIF11 in breast cancer, it may be worth exploring the function of FBXO30-KIF11 interactions in cancers [125]. Moreover, it has been confirmed that tripartite motif 8 (TRIM8) interacts with KIF11, and the downregulation of TRIM8 leads to a delay in mitosis progression. However, the mechanism of ubiquitination is still unclear [162].

## 5. Oncogenic Functions of KIF11

With an accumulating number of studies reporting the dysregulation mechanisms of KIF11 in cancers, significant investigations have been conducted to elucidate the oncogenic properties of this kinesin. Various oncogenic signaling pathways and several protein partners are driven by KIF11, promoting important hallmarks of cancers, including cancer cell proliferation, invasion, and metastasis (Figure 4).

### 5.1. KIF11 in Cancer Proliferation and Survival

In relation to cancer, KIF11 has been found to support cancer proliferation via the Wnt/β-catenin pathway. In breast cancer, knocked down KIF11 significantly reduces the level of cancer stem cell markers and number and size of breast cancers in vivo through inactivating the Wnt signaling pathway via inhibiting the nuclear translocation of β-catenin and downregulating the expression of transcription factor T-cell factor 1 (TCF1) and cyclin D1 (CCND1), the known downstream gene of this pathway [163]. In HCC cells, KIF11 is targeted by ASPM, and overexpressed KIF11 restores the inhibiting effect on the expression of β-catenin and phosphorylated GSK-3β, which result from the knockdown of ASPM, showing the upregulation of the Wnt signaling pathway and the promotion of proliferation-associated proteins [164]. Since the involvement of ASPM in the Wnt signaling pathway has been reported in several cancers [165,166], the mediation of KIF11 throughout this mechanism should be explored in different cancers to give insights for targeted drug development. Additionally, upregulated KIF11 enhances the membrane trafficking of human epidermal growth factor receptor 2 (ERBB2) and activates the PI3K/AKT signaling pathway to enhance the proliferation of GBC cells [27].

Another mechanism contributing to cancer proliferation has been reported in pancreatic ductal adenocarcinoma (PDAC). The mevalonate metabolism pathway is involved in multiple cellular functions, including cholesterol synthesis, playing a vital role in tumor proliferation [167]. The Cancer Genome Atlas (TCGA) reveals a strong correlation between KIF11 expression and the mevalonate metabolism pathway, which is regulated by the sterol regulatory element-binding protein 2 (SREBP2) protein. Upregulated KIF11 attenuates the ubiquitination of SREBP2, leading to the stabilization of SREBP2 and an increase in the free cholestenone concentration, thereby promoting PDCA cell growth and colony formation [101]. Interestingly, KIF11 interacts with PFKM, which is a regulator of glycolysis [127]. Hence, these studies provide insights into the involvement of KIF11 in tumor cell metabolism, which is another essential hallmark of cancer that changes the microenvironment for progression.

### 5.2. KIF11 as a Promotor of Cancer Invasion and Metastasis

#### 5.2.1. KIF11 in Epithelial–Mesenchymal Transition (EMT)

A number of studies have documented that KIF11 promotes the invasion and migration of cancer cells [101,110,168] and is a regulator of the EMT program, which is a process whereby cells gradually change their epithelial properties into mesenchymal-like cells, breaking the cell–cell adhesion to gain the ability to metastasize [169]. In clear cell renal cell carcinoma, inhibiting KIF11 increases the expression level of E-cadherin mRNA and decreases the mRNA level of mesenchymal markers, including N-cadherin and vimentin [118]. Similar results have been observed in ovarian cancer with an increasing level of ZEB-1, a transcriptional repressor of E-cadherin expression, instead of upregulated vimentin expression [134]. Apart from the E-cadherin/N-cadherin phenotypic switch, KIF11 can also regulate the expression of extrinsic factors to play roles in cell migration. In glioma, using KIF11 inhibitors significantly inhibits the expression of four types of matrix metalloproteinases (MMPs), including MMP-1/2/3/9, causing extracellular matrix turnover and cancer cell migration [126,170]. Though the mechanisms of the KIF11-mediated E-cadherin/N-cadherin phenotypic switch and MMP alternation have not been identified, the regulation of both extrinsic and intrinsic factors in remodeling the extracellular matrix remains crucial for the pro-metastatic function of KIF11.

Regulated by KIF11 and its interacting partners, different cell signaling pathways have been reported to promote the EMT process and enhance their invasive and migratory capabilities. In breast cancer, suppressing PI3K/AKT and MAPK/ERK might partially explain the cadherin switch mediated by the downregulation of KIF11 [106]. Furthermore, confirmed by a rescue experiment conducted on ovarian cancer cells, the proteins associated with the migration of these two pathways are activated by a complex combining death receptor 6 (DR6), TRAF4, and KIF11 [171]. The Wnt/β-catenin pathway is also a well-known oncogenic signaling pathway, and it is activated by the aforementioned ASPM/KIF11 axis to participate in invasion, migration, and the EMT program [32,164]. Additionally, Xue T et al. claimed that downregulated KIF11 represses the expression of phosphorated p65 (NF-κB) and phosphorated JNK to be the upstream regulator of the NF-κB/JNK signaling pathway, inhibiting NSCLC cell migration [22]. However, the results showed that KIF11 silencing could rescue CVB-D-mediated cancer growth inhibition, though they both inhibited the NF-ΚB/JNK pathway, making this conclusion debatable.

#### 5.2.2. KIF11 in Tumor Angiogenesis

KIF11-mediated angiogenesis has been identified in several cancers and occurs mainly through KIF11 interacting with the vascular endothelial growth factor (VEGF), an angiogenic inducer that participates in solid tumors growth. For example, KIF11 is significantly more upregulated in PCa tissues during metastasis compared with non-metastasis tissues [105,123]. Since KIF11 expression is positively corelated with VEGF, angiogenesis might be mediated by KIF11 through it interacting with VEGF to promote the occurrence of bone metastasis [105,172]. However, contrary conclusions have been made in previous studies concerning the regulation patterns between KIF11 and VEGF, representing a topic for further exploration. Doan et al. suggested that compared with suppressing KIF11 or VEGF separately, using an siRNA cocktail to downregulate KIF11 and VEGF simultaneously remarkably inhibits HCC cell migration and invasion. It is worth noting that the knockdown of VEGF can decrease the expression level of KIF11, presenting the regulatory effect of VEGF to KIF11 [168]. However, a significantly decreased VEGF expression level in gastric adenocarcinoma has been observed after inhibiting the expression of KIF11 [173]. Furthermore, KIF11 may modulate VEGF through the PI3K/AKT and ERK1/2 pathways, participating in vascular formation, but further research is required to elucidate this mechanism [173].

## 6. Overview of KIF11 Inhibitors

Studies on KIF11 controlling mitosis provide insights into novel drug exploration to disrupt the expression of KIF11 during chemotherapy. The first inhibitor of KIF11 to be reported was monastrol, which blocks the release of ADP and changes the structure of the spindle [174]. Nevertheless, monastrol’s weak activity and significant adverse effects restrict its application, promoting modified structure studies and other types of inhibitors. According to the acting mechanism, KIF11 inhibitors are divided into loop 5-binding allosteric inhibitors (e.g., ispinesib, S-Trityl-L-cysteine (STLC), and Arry-520) and competitive ATP-binding inhibitors (e.g., BRD9876, GSK1, and PVZB1194). A thorough discussion of the chemical classes of KIF11 inhibitors can be seen in the review of El-Nassan et al. [175]. Currently, approximately 10 types of KIF11 inhibitors have been used in clinical trials (Table 3). The phase I trials provided no neurotoxicity data, as expected, and only common adverse effects happened at the maximal tolerated dose, such as neutropenia and hepatotoxicity [36]. However, the existing inhibitors have not shown satisfactory effectiveness in several types of cancers. For example, ispinesib administration did not cause a treatment reaction in patients with melanoma, prostate cancer, and squamous cell carcinoma of the head and neck [176,177,178]. Though Arry-520 is a promising agent used to treat patients with relapsed or refractory multiple myeloma, no phase III trials of any KIF11 inhibitor have been approved by the FDA [179]. The point mutation of loop 5 in tumor cells contributes to the resistance of KIF11 inhibitors. For example, the mutation of T107N, which is in the P-loop of the ATP-binding site is associated with resistance to STLC [180]. When KIF11 is absent, KIF15 (another member of the kinesin superfamily) can construct a complete bipolar spindle, leading to the proliferation of tumor cells that escape repression from the KIF11 inhibitors [181]. Other results have demonstrated that inhibiting KIF15 hinders tumor cells from developing resistance to KIF11 inhibitors [182]. The simultaneous knockdown of KIF11 and KIF15 represses the proliferation of GC cells, providing an insight into an effective combination therapy strategy [183].

## 7. Future Directions

Given the central role of KIF11 dynamics in controlling mitosis, it is crucial to investigate the regulation systems consisting of KIF11 and the carried cargos. Ensconsin is a microtubular-associated protein that regulates kinesin-1 through highly specific mechanisms [207,208]. TPX2 is another kind of microtubular-associated protein, but its control of KIF11 cannot be found on kinesin-1 [96]. Therefore, the regulation of TPX2 on KIF11 might represent a novel mechanism that is worth exploring. Several KIF11 interacting proteins have been mentioned, but their mechanisms and a complete protein–protein interaction network need to be established. Weinger et al. indicated that the tail domain of KIF11 encourages binding between KIF11 and the MTs, implicating the existing interactions between the three domains [70]. Therefore, it is important to provide clear evidence, and the disruption of these interactions perhaps represent a novel approach to the study of new drugs. Additionally, the bi-directional movement of yeast KIF11 has been mentioned [68,76,77]. Its approach and significance remain burning issues. If the minus-end-directed motion can be identified in other KIF11 homologs, novel movement patterns will become the new hotspot of kinesin research, strengthening the understanding of KIF11 functions.

Recently, the functions, expressions, and clinical significance of KIF11 in multiple cancers have been demonstrated. The mevalonate metabolic pathway, which is known for its capacity to induce immune responses, has been demonstrated to have a close association with KIF11 expression. When combined with other investigations exploring the relationship between aberrant KIF11 expression and immune cell infiltration, these studies offer valuable insights into the roles of KIF11 in the cancer microenvironment. Furthermore, the interaction between metabolic and cellular transduction pathways has been widely explored [209]. Thus, studies on the KIF11 in signaling pathway regulation in cancer will continue to shed light on its involvement.

More profound research on targeting inhibitors with fewer side effects for monotherapy or combination therapy in cancer treatment is needed. Additionally, the non-mitotic roles of KIF11 in neurons may give insights into the role of KIF11 inhibition in the elongation of axons and cancer pain regulation. L. Ferhat was the first to observe the expression of KIF11 in the postmitotic neurons of rodents [40]. During neuron development, the KIF11 protein quantity decreases, but it still exists in adult neurons both in the CNS and PNS [41]. KIF11 generates a force to regulate MT organization in the developing axons, acting as a brake to stabilize short MTs and restrict axon growth [41,210]. Using monastrol, which is a kind of KIF11 inhibitor, significantly elongates the axons’ lengths. Mechanically, monastrol inhibits the retraction of neurons, resulting in an enhanced branching frequency [43,210]. Therefore, inhibiting KIF11 may become an effective therapeutic strategy for people with injured nervous systems. Furthermore, inhibiting KIF11 modulates synaptic transmission and inhibits thermal pain [211,212]. However, Chen et al. demonstrated that KIF11 promotes the membrane translocation of NKCC1 in the spinal cord, which is similar to the mechanism of paclitaxel-induced neuropathic pain [213]. Therefore, further explorations of the pain modulation network regulated by KIF11 will help to guide the development of cancer treatments.

## Figures and Tables

**Figure 1 biomolecules-14-00386-f001:**
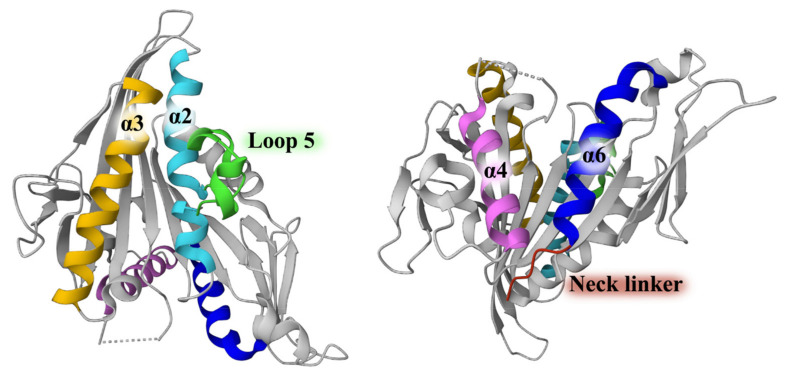
Two views of KIF11 motor domain (PDB ID: 1YRS). The structures highlighted on the left, representing the binding sites of allosteric inhibitors, are loop 5 (green), α2 (light blue), and α3 (yellow); the structures highlighted on the right represent the binding sites of ATP competitive binding inhibitors: neck linker (red), α4 (pink), and α6 (dark blue).

**Figure 2 biomolecules-14-00386-f002:**
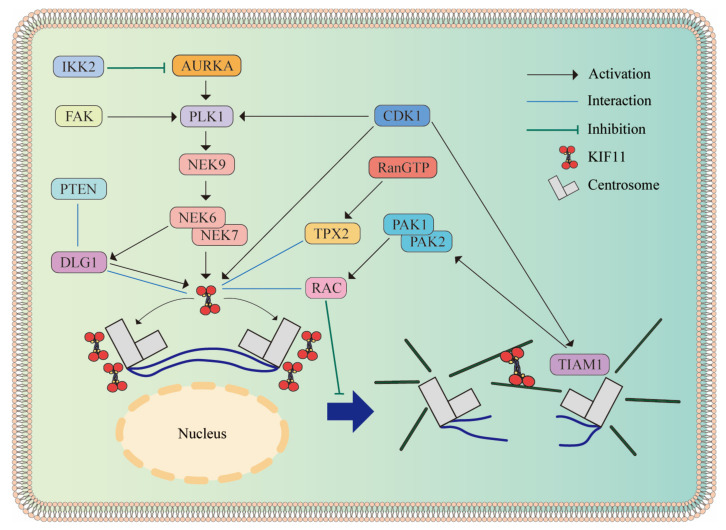
Molecules involved in KIF11 accumulation, controlling centrosome separation. CDK1 plays a crucial role in the centrosome localization of KIF11. It not only directly influences KIF11 but also collaborates with PLK1, which is modulated by AURKA, initiating the NEK6/7/9 cascade. This cascade, in turn, governs the accumulation of DLG1 and KIF11. The activation of TIAM1 by CDK1 drives PAK1/2 to act on RAC, which interacts with KIF11 to inhibit centrosome separation. Under the regulation of RanGTP, TPX2 is another protein that binds to KIF11. Neither activation nor inhibition, the light blue line displays other interactions between two molecules, both of which form a complex to execute functions.

**Figure 3 biomolecules-14-00386-f003:**
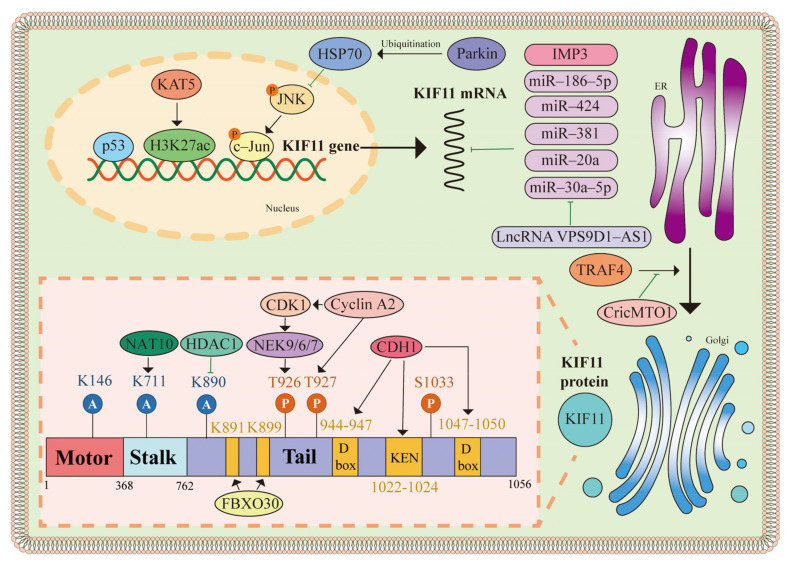
Regulation of abnormal KIF11expression. KIF11 transcription is regulated by transcription factors, histone modification, and the JNK/c-Jun pathway. Non-coding RNA, including miR-185-5p, miR-424, miR-381, miR-20a, and miR-20a-5p, can regulate the mRNA level of KIF11. IMP3 plays a role in this process as well. A schematic representation of the post-translational regulation is framed by the orange box. KIF11 comprises a motor domain (in red), a stalk domain (in light blue), and a tail domain (in purple). Phosphorylation sites are denoted by orange circles. Acetylation sites are denoted by blue circles, and ubiquitination sites are denoted by yellow blocks.

**Figure 4 biomolecules-14-00386-f004:**
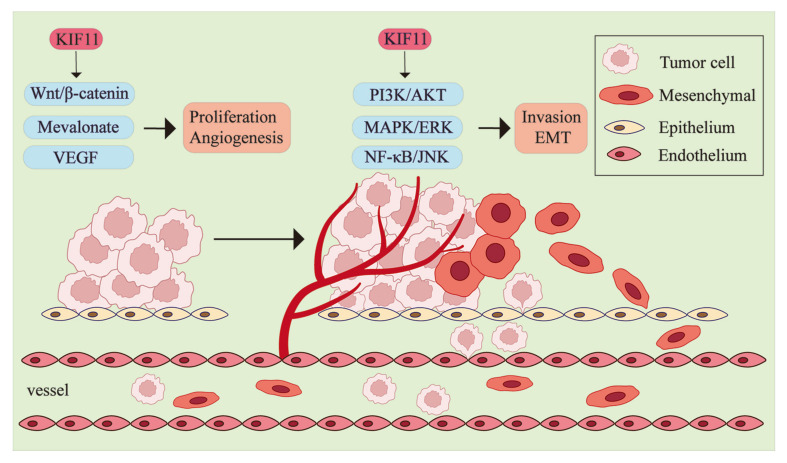
Regulation of KIF11 in various oncogenic signaling pathways. This figure is a schematic representation of the epithelial mesenchymal transition (EMT), angiogenesis, and cancer cell proliferation, invasion, metastasis. KIF11 promotes the activation of the Wnt/β-catenin and mevalonate metabolism pathways to enhance cancer cell proliferation and interacts with VEGF to promote angiogenesis. Additionally, KIF11 is involved in the PI3K/AKT, MAPK/ERK, and NF-κB/JNK pathways, promoting cancer cell invasion and the EMT process.

**Table 2 biomolecules-14-00386-t002:** Regulation and oncogenic functions of KIF11.

Cancer Types	Models	Pathways	Functions	Ref.
Glioma	Glioblastoma (GBM) cell lines	\	Inhibiting KIF11 promotes cancer cell apoptosis and represses the epithelial–mesenchymal transition (EMT) process to decrease tumor spreading.	[126]
	GBM cell lines, xenograft-bearing mice	\	Upregulated KIF11 promotes motility and changes morphogenesis by lengthening microtubules of glioblastoma tumor-initiating cells (TICs). And inhibiting KIF11 decreases TICs’ self-renewal.	[17]
	GBM cell lines	PFKM/KIF11	PFKM stabilizes KIF11 to promote GBM proliferation and invasion.	[127]
Head and neck squamous cell carcinoma (HNSCC)	HNSCC cell lines	\	KIF11 is upregulated in HNCSS cells, and it is important for cell viability.	[128]
Thymic epithelial tumor (TET)	TET specimens	\	KIF11 is upregulated in TET tissues and is significantly upregulated in thymic carcinoma compared to thymoma.	[129]
Malignant peripheral nerve sheath tumors (MPNST)	MPNST cell lines	\	Knockdown of KIF11 decreases the cell viability of MPNST cells.	[130]
Meningioma	Meningioma cell lines, specimens	\	Knockdown of KIF11 inhibits meningioma cell proliferation.	[113]
Non-small-cell lung cancer (NSCLC)	NSCLC cell lines	CVB-D/NF-ΚB/JNK/KIF11-CDC25C-CDK1-cyclinB1	KIF11 acts as the regulator of the NF-ΚB/JNK pathway and a member of the KIF11-CSC25C-CDK1-cyclinB1 G2/M phase transition regulatory network and is inhibited by CVB-D to suppress the growth and progression of NSCLC cells.	[22]
Lung adenocarcinoma (LUAD)	LUAD specimens	EGFR_1/SLC2A1-CCNB2-HMMR-KIF11-NUSAP1-PRC1-UBE2C	KIF11 acts as a member of the SLC2A1-CCNB2-HMMR-KIF11-NUSAP1-PRC1-UBE2C axis and is activated by EGFR_1 to promote the mobility of LUAD cells.	[131]
	LUAD cell lines	\	Downregulated KIF11 inhibits cell proliferation and increases cell proportion in the G2/M phase and inhibits cell migration and invasion.	[110]
Hepatocellular carcinoma (HCC)	HCC specimens, cell lines, xenograft-bearing mice	\	Inhibiting KIF11 reduces cell viability, disrupts the cell cycle, induces cell apoptosis, and decreases tumor growth in vivo.	[23]
Colorectal cancer (CRC)	Colon cancer cell lines	Cyclin A2/KIF11	Depletion of cyclin A2 decreases the phosphorylation of KIF11 at T926. And overexpressed cyclin A2 induces centrosome amplification, which may be mediated by KIF11.	[132]
	CRC cell lines, xenograft-bearing mice	\	Depletion of KIF11 decreases CRC growth in vitro and in vivo.	[19]
	CRC specimens, cell lines	\	KIF11 is important for spheroid formation by CRC cells.	[107]
Esophageal cancer (ESCA)	ESCA specimens, cell lines	\	KIF11 is important for spheroid formation by ESCC cells.	[107]
Ovarian cancer	Ovarian cancer specimens	MiR-424/KIF11, miR-381/KIF11	Identification of candidate biomarkers and analysis of prognostic values in ovarian cancer by integrated bioinformatics analysis	[133]
	Ovarian cancer cell lines	\	KIF11 is upregulated in ovarian cancer and promotes ovarian cancer cell proliferation and invasion.	[134]
Prostate cancer (PC)	PC cell lines	MiR-429-ANLN-KIF11	KIF11 acts as a member of the miR-429-ANLN-KIF11 pathway to serve as a novel prognostic biomarker.	[135]

**Table 3 biomolecules-14-00386-t003:** KIF11 inhibitors in clinical trials.

Chemical Class	Name/Structure	Company	Phase	Condition	Clinical Trial	Ref.
Pyrrole	MK-0731	Merck Sharp & Dohme LLC	P1	Solid	NCT00104364/completed.	[184]
Thiadiazole	Filanesib (Arry-520)	Array Biopharma	P1, P2	Solid, MM, R/R PCM, AML, MDS	NCT02384083/completed; NCT01372540/completed; NCT00637052/completed; NCT00462358/completed; NCT00821249/completed; NCT02092922/completed; NCT01248923/completed; NCT01989325/completed.	[36,185,186,187,188,189]
Quanizoles	Ispinesib (SB-715992)	National Cancer Institute, GlaxoSmithKline, Cytokinetics, NCIC Clinical Trials Group	P1, P2	Solid, RCC, BC, HNSCC, lymphoma, OC, MDS, PC, NSCLC, melanoma, LC, CRC,	NCT00354250/completed; NCT00089973/completed; NCT00119171/completed; NCT00095628/completed; NCT00607841/terminated; NCT00363272/completed; NCT00097409/completed; NCT00098826/completed; NCT00136578/completed; NCT00101244/terminated; NCT00096499/completed; NCT00085813/completed; NCT00095953/completed; NCT00095992/completed; NCT00103311/completed; NCT00169520/completed.	[38,39,176,177,178,190,191,192,193,194]
Quanizoles	SB-743921	Cytokinetics	P1, P2	Solid, NHL, HL	NCT00343564/completed; NCT00136513/completed.	[195,196]
Thiazoles	LY2523355	Kyowa Kirin Co., Ltd., Eli Lilly	P1, P2	Solid, BC, OC, NSCLC, RC, GEC, HNSCC, AC, SCLC	NCT01358019/completed; NCT01416389/completed; NCT01214655/terminated; NCT01059643/completed; NCT01214629/completed; NCT01214642/completed; NCT01025284/completed.	[197,198]
Thiazolopyrimidines	AZD4877	AstraZeneca	P1, P2	NHL, Bladder cancer, AML,	NCT00471367/terminated; NCT00661609/completed; NCT00613652/completed; NCT00486265/terminated; NCT00389389/completed; NCT00460460/terminated.	[199,200,201,202,203]
Quanizoles	ARQ621	ArQule	P1	Solid	NCT00825487/completed.	[204,205]
siRNA	ALN-VSP02	Alnylam Pharmaceuticals	P1	Solid,	NCT01158079/completed; NCT00882180/completed.	[206]

Abbreviation: MM, multiple myeloma; R/R PCM, recurrent/refractory plasma cell leukemia; AML, acute myeloid leukemia; MDS, myelodysplastic syndrome; RCC, renal cell cancer; BC, breast cancer; HNSCC, head and neck squamous cell carcinoma; OC, ovarian cancer; PC, prostate cancer; NSCLC, non-small-cell lung cancer; LC, liver cancer; CRC, colorectal cancer; NHL, non-Hodgkin’s lymphoma; HL, Hodgkin’s lymphoma; GEC, gastroesophageal cancer; AC, advanced cancer; SCLC, small-cell lung cancer.

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
