# Peer review of "Mitotic Functions and Characters of KIF11 in Cancers"

_biomolecules, 2024, doi:10.3390/biom14040386_

Round 1

Reviewer 1 Report

Comments and Suggestions for Authors

A comprehensive review of the functions of KIF11 and the clinical importance.

There are not significant issues. I would recommend having an independent section about KIF11 inhibitors as anti-cancer drugs. 

It is very difficult to read the tables in the current format. Cannot tell which word belongs to which line. 

Reviewer 2 Report

Comments and Suggestions for Authors

Comments on the Quality of English Language

Reviewer 3 Report

Comments and Suggestions for Authors

In the manuscript entitled “Mitotic functions and characters of KIF11 in Cancers” the authors review the literature focused in the mitotic kinesin KIF11, and its potential roles in cancer. The manuscript discusses the mitotic functions of KIF11 and the multiple interacting proteins and kinases that regulate KIF11 activity during spindle pole separation and mitotic spindle assembly. With the discovery of the first non-competitive inhibitor of KIF11 (Monastrol), KIF11 became an attractive candidate for chemotherapeutic intervention (and an alternative to taxanes). The authors nicely summarized the classes of inhibitors and the clinical studies involving these inhibitors. Most of the review discusses the potential clinical significance of Kif11 dysregulation: as a biomarker for poor outcomes, and the correlation between KIF11 and a variety of processes associated tumorigenesis, tumor angiogenesis and metastasis.

The literature review is quite comprehensive, and reflects the attention garnered by this motor. If there were any omissions, it was that the manuscript did not mention that the literature on Kif11-independent spindle assembly, which occurs through the actions of Kif15. Kif15 overexpression can represent a mechanism for resistance to Kif11 inhibitors, and thus is relevant for this discussion. And while it is more of a comment, a fair amount of the discussion is committed to correlative studies tying Kif11 to different signaling pathways and cancer processes, most of which have no molecular mechanism tying the actions of a plus-ended, bipolar homo-tetrameric microtubule motor to these largely non-mitotic phenomena. The authors might consider some caution when placing weight on these purely correlative studies.

Minor point:

·      Throughout the manuscript there are grammatical errors that are minor but are too numerous. In some cases, these obscure the meaning of the sentence and thus diminish the impact of the point the authors are trying to make.

·      As mentioned above, there is a strong emphasis on largely correlative studies of Kif11 during cancer, but in terms of non-mitotic roles of Kif11, its roles in neuronal morphogenesis is perhaps the most well-established. The authors do mention that Kif11 drugs do have effects on memory, but do not expand on this any further. The authors might look at that literature and see if there are any reports that might shed any light on Kif11’s potential roles in cancer.

Comments on the Quality of English Language

There are minor grammatical errors throughout the manuscript.
